# *In vitro* transcription accurately predicts lac repressor phenotype *in vivo* in *Escherichia coli*

Matthew Almond Sochor

Department of Biochemistry and Biophysics, Perelman School of Medicine, University of Pennsylvania, Philadelphia, PA, USA

## ABSTRACT

A multitude of studies have looked at the *in vivo* and *in vitro* behavior of the lac repressor binding to DNA and effector molecules in order to study transcriptional repression, however these studies are not always reconcilable. Here we use *in vitro* transcription to directly mimic the *in vivo* system in order to build a self consistent set of experiments to directly compare *in vivo* and *in vitro* genetic repression. A thermodynamic model of the lac repressor binding to operator DNA and effector is used to link DNA occupancy to either normalized *in vitro* mRNA product or normalized *in vivo* fluorescence of a regulated gene, YFP. An accurate measurement of repressor, DNA and effector concentrations were made both *in vivo* and *in vitro* allowing for direct modeling of the entire thermodynamic equilibrium. *In vivo* repression profiles are accurately predicted from the given *in vitro* parameters when molecular crowding is considered. Interestingly, our measured repressor–operator DNA affinity differs significantly from previous *in vitro* measurements. The literature values are unable to replicate *in vivo* binding data. We therefore conclude that the repressor-DNA affinity is much weaker than previously thought. This finding would suggest that *in vitro* techniques that are specifically designed to mimic the *in vivo* process may be necessary to replicate the native system.

## INTRODUCTION

The lac genetic switch consists of the lac repressor, a short "operator" DNA sequence, and effector molecules (*Swint-Kruse & Matthews, 2009*). The minimal functional lac repressor is homodimeric and includes an N-terminal DNA binding domain and two effector binding sites (one per monomer). Repressor binds to operator DNA preventing RNA polymerase from transcribing downstream genes. Effector molecules bind to each effector binding site causing an allosteric transition wherein repressor dissociates from operator DNA allowing transcription to proceed (*Lewis, 2005*). Previously our lab has used a standard Monod, Wyman, and Changeux (MWC) model of thermodynamic equilibrium to model the behavior of the lac genetic switch (Fig. 1) (*Monod, Wyman & Changeux, 1965*). The MWC model considers two structural conformations of the lac repressor,

Corresponding author
Matthew Almond Sochor,
msochor@mail.med.upenn.edu

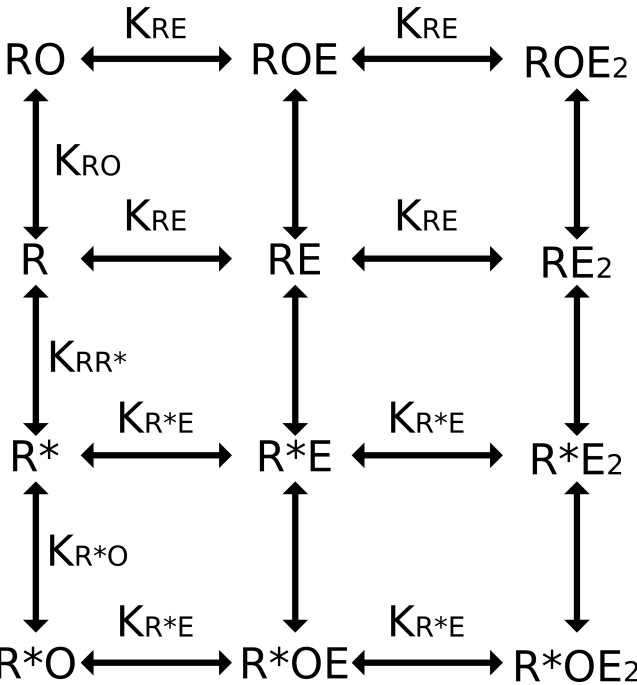

**Figure 1 Monod, Wyman, and Changeux (MWC) model of thermodynamic equilibrium.** This model identifies two primary structural conformations of the lac repressor (*R* and *R\**): the *R* state has high operator DNA (*O*) affinity and the *R\** state has low operator DNA affinity. Addition of effector (*E*) alters the effective equilibrium between the two states allowing for an increase or decrease in amount of operator DNA bound. Fraction of bound operator was considered a proxy for transcription; unbound operator can be freely transcribed. Thermodynamic binding and conformational equilibrium constants are fully defined in the methods.

defined as *R* and *R\**, which each have unique affinities for the ligands (DNA and effector) and are in equilibrium.

While the underpinnings of the lac genetic switch have been well characterized, it is less well understood how to utilize this information to achieve practical goals. How do we reduce the background leakiness of the repressor? Can you do so without compromising maximal inducibility? Can you target certain phenotypic properties through directed mutation? Will novel genetic switches developed in *E. coli* perform the same in different cell types? Significant advancement has been made in recent years towards answering these more complex questions.

*Daber, Sharp & Lewis (2009)* examined the number of effector molecules necessary to induce transcription. Heterodimeric lac repressors were created that bound either 0, 1 or 2 effector molecules and the *in vivo* regulation of a fluorescent gene was measured. An analytical solution of a simplified MWC equilibria allowed for direct measurements of dimensionless bulk parameters comprised of combinations of thermodynamic binding constants and species concentrations. While these parameters were useful in showing that two effector molecules are required for fully inducing the genetic switch, they were unable to measure the thermodynamic constants themselves.

*Daber, Sochor & Lewis (2011)* next sought to link distinct perturbations of the lac genetic switch to changes in thermodynamic parameters. Mutations were made in the DNA binding domain and effector binding pocket of the repressor. They were able to measure the repressor-effector binding affinities; however they still could only measure a dimensionless constant which contained repressor concentration and repressor-DNA affinity. Mutations in the DNA binding domain of the lac repressor were linked to changes in the repressor-DNA affinity. Alternatively, changes in the repressor concentration could also account for the phenotype. Mutations in the effector binding domain did alter the effector binding affinities. Interestingly, effector binding domain mutations were also linked to changes in the conformational equilibrium of the repressor, but once again changes in the repressor concentration could account for the phenotype. These results were encouraging evidence that directed mutations lead to directed phenotypes, but the question of repressor concentration clouded the picture.

A study by *Poelwijk, de Vos & Tans (2011)* looked for unique phenotypes through random mutagenesis of the lac repressor. Mutants were identified which exhibited an inverted repression behavior; a phenotype also found by *Daber, Sochor & Lewis (2011)* by mutating the effector binding domain. Interestingly, Poelwijk's mutations were in regions physically distinct from either the DNA or effector binding domains. One potential explanation was that the mutations destabilize the folded form of the repressor, altering the conformational landscape. Mutagenesis of the repressor can result in more than just predictable changes of thermodynamic binding constants.

Central to all of these studies was the use of *in vivo* data to understand the behavior of genetic switches. It has been pointed out that a lack of corroborating *in vitro* evidence prevents the identification of other processes which may significantly play into the equilibrium, such as non-specific DNA binding or effector uptake (*Tungtur et al., 2011*). They attempted to measure the thermodynamic binding of a LacI/GalR hybrid repressor both *in vitro* and *in vivo*. Notably, a DNA pull down assay was used to quantify the *in vivo* concentration of their hybrid repressor. Unfortunately, they were unable to rectify a greater than 25-fold difference between their two data sets. This indicates that they are missing a significant contributor to the genetic switch by only analyzing *in vivo* data.

Here we sought to overcome the limitations of past studies three ways: (1) measure the *in vivo* concentration of the lac repressor, (2) measure the *in vitro* transcription of purified lac genetic switch, and (3) use an assumption free solution to the MWC equilibrium to model both *in vitro* and *in vivo* data.

We were able to measure lac repressor concentration *in vivo* and use *in vitro* transcription to assess the purified lac genetic switch. Furthermore, we found excellent agreement between *in vitro* and *in vivo* data when molecular crowding was taken into consideration. We do however find that the repressor-DNA affinity was much lower than has previously been measured *in vitro*. Additional concerns, such as effector uptake and non-specific DNA binding do not appear to play significant roles.

## MATERIALS AND METHODS

### *In vivo* measurement of lac genetic switch

Reporter plasmid was made as previously reported (*Daber & Lewis, 2009*) with the O1 operator sequence (5′-AA TT GTG AGC G GAT AAC AA TT-3′) followed by YFP and providing ampicillin (AMP) resistance. Lac repressor was expressed on a second plasmid as previously described (*Daber & Lewis, 2009*) providing chloramphenicol (CAM) resistance. A C-terminal mCherry tag was added to the Lac repressor gene after an 11bp linker to create the Lac-mCherry construct.

We double transformed reporter and repressor plasmids into EPB229 cells ($F^-\Delta$(lacI-lacA)::frt). These cells were derived from the MG1655 "wild type" line. Colonies were picked in triplicate into MOPS minimal media with 0.4% glucose, AMP and CAM and grown overnight at 37 °C with shaking. 50 μL of the overnight culture was used to inoculate 1 mL fresh MOPS minimal media supplanted with varying amounts of IPTG. We measured optical density at 600 nm ($OD_{600}$), YFP fluorescence (excite: 510 nm emit: 535 nm), and mCherry fluorescence (excite: 585 nm emit: 610 nm) for all wells at 1 h intervals over a 12 h period using a TECAN M1000 plate reader in 384 well optical bottom plates (Corning).

### Purification of Lac-mCherry

Lac-mCherry was cloned into the pBAD-DEST49 expression vector (Clontech). A 6xHis C-terminal tag was added to aid in purification. BL21(DE3) cells were transformed and grown to mid-log at 37 °C with shaking in 2xYT media. At mid-log growth, expression of Lac-mCherry was induced with the addition of arabinose 0.1% (v/v) and the temperature was reduced to 15 °C and cells were allowed to grow overnight (approximately 12–16 h). Cell extract was purified with Ni-NTA beads (Clontech) and a sizing column (HiLoad 16/60 Superdex 75 Prep Grade with AKTA Prime FPLC) and purified Lac-mCherry was equilibrated into GF buffer (200 mM Tris pH 7.4, 200 mM KCl, 10 mM EDTA, 3 mM DTT).

### Native gel electrophoresis

We used the NativePage (Invitrogen) kit for native gel electrophoresis. The primary advantage of this kit is that it is based upon the Blue Native Polyacrylamide Gel Electrophoresis (BN Page) which uses Coomassie G-250 as the molecule to provide charge shift for proteins. Coomassie G-250 binds to proteins providing a net negative charge without denaturing the protein.

Purified Lac-mCherry protein was thawed and mixed in non-denaturing sample loading buffer (final concentration of Lac-mCherry = 500 nM) and Native Mark (Invitrogen) protein standard were added to wells of NativePage 4%–16% Bis-Tris gels. The gel was loaded into a Novex Mini-Cell (Invitrogen) gel running box.

The interior chamber was filled with NativePage cathod running buffer which contains Coomassie G-250. The exterior was filled with NativePage anode running buffer. The gel was run at 150 V for 2 h. The gel was removed and placed in Fix (40% methanol, 10% acetic

acid), microwaved on high for 45 s, and shaken on an orbital shaker for 15 min to fix. The gel was then placed in Destain (8% acetic acid), microwaved on high for 45 s, and placed on an orbital shaker overnight to remove unbound Coomassie G-250. Gel was imaged with white light to visualize Lac-mCherry oligomerization state.

## Measuring *in vivo* concentrations of the lac repressor

EPB229 cells were co-transformed with Lac-mCherry and O1 YFP reporter. An individual colony was picked into MOPS minimal media with 0.4% glucose, AMP and CAM and grown overnight at 37 °C with shaking. 50 μL was innoculated into 1 mL fresh media and grown to mid-log phase.

Purified Lac-mCherry was quantified with both a BCA Assay Kit (Pierce) and optical $A_{280}$ measurements using a NanoDrop 2000 Spectrometer (Thermo Scientific). Dilutions were made over 8 orders of magnitude and 50 μL was loaded into clear bottom 384 well plates in triplicate. mCherry fluorescent measurements (excite: 585 nm emit: 610 nm) were made using various gains to establish linear regimes for the instrument (TECAN M1000).

We established a raw cell count by plating dilutions of a culture of EPB229 cells. Serial dilutions were made over 10 orders of magnitude and each dilution had $OD_{600}$ measured (TECAN M1000 and Ultrospec 2100 pro) and 100 μL plated onto LB agar with AMP and CAM. We found $1.92 \times 10^6$ cells/μL at mid-log growth phase which was about two-fold higher than standard estimates of $1 \times 10^6$ cells/μL for *E. coli*. Aliquots of known cell counts were then used to establish a linear relationship with $OD_{600}$ on our plate reader. Similarly, purified Lac-mCherry of known concentration was used to establish a linear relationship with mCherry fluorescence on our plate reader at a fixed gain.

EPB229 cells were co-transformed with plasmid constitutively expressing Lac-mCherry and a reporter plasmid which has YFP under the control of the natural operator O1. We measured mCherry fluorescence at a fixed gain and $OD_{600}$ from which we calculated the concentration of Lac-mCherry in the well and the number of cells in the well. The approximate volume of *E. coli* was estimated to be $1 \times 10^{-15}$ L (*Kubitschek & Friske, 1986*). Multiplying volume of *E. coli* by number of cells allows us to estimate what fraction of the well volume was intracellular.

Calibration of raw mCherry fluorescent signal and $OD_{600}$ was converted to intracellular repressor concentration.

## Fluorescent data processing

*In vivo* data was normalized for growth by measuring cells in triplicate as they were growing. All data points collected were then fit to a 2nd order polynomial to obtain a curve which was fluorescence as a function of $OD_{600}$ and error of these fit is reported as the error bars of the *in vivo* fluorescence. Positive control was established by co-transforming EPB229 cells with O1 YFP reporter and a CAM plasmid without Lac-mCherry (pABD34). YFP signal was normalized to the polynomial fit from the positive control. Final values for fitting were calculated for cells at approximately mid-log growth phase ($OD_{600} = 0.4$).

## Measuring *in vitro* transcription

A reporter plasmid was made with the O1 operator after a T7 promoter. Reporter was linearized to 450bp and purified by spin column purification (Clontech).

MaxiScript T7 kit (Ambion) was used to perform *in vitro* transcription. $CTP[\alpha\text{-}^{32}P]$ was incorporated into mRNA transcripts and the water fraction of the standard reaction was supplanted with varying concentrations of Lac-mCherry and IPTG. Final buffer conditions were 6 μg linearized template DNA (5.5 μM total DNA concentration); 20 mM Tris–HCl, pH 7.9; 20 mM MgCl2; 1 mM dithiothreitol (DTT); 2 mM spermidine-HcL; 0.5 mM GTP, ATP, and UTP; 0.25 mM CTP; and 2.5 μM $CTP[\alpha\text{–}^{32}P]$. Transcription was allowed to proceed for 30 min at 37 °C until halted by boiling. Samples were loaded onto polyacrylamide gels and electrophoresis was used to separate free $CTP[\alpha\text{–}^{32}P]$ from that incorporated into mRNA. Gels were dried and exposed to radiological plates. Plates were imaged on a Typhoon scanner and bands were quantified using ImageJ (NIH). All experiments were performed *in vitro* transcription experiments were performed in duplicate on different days. Reported values are averages of the trials and error bars are standard deviations.

## Modeling

Experimentally, we would like to measure the output from a promoter regulated by the lac genetic switch. It was assumed that transcription by RNA polymerase from the promoter was linearly related to the occupancy of the DNA operator within the promoter by the lac repressor,

$$\text{transcription} \propto \frac{[O]}{[O]_{\text{tot}}}. \tag{1}$$

In order to model experimental data, we need to compute the occupancy of the DNA operator in terms of the thermodynamic constants ($K_{RR^*}$, $K_{RE}$, $K_{R^*E}$, $K_{RO}$, and $K_{R^*O}$) and the total concentration of repressor, effector and operator ($[R]_{\text{tot}}$, $[E]_{\text{tot}}$, and $[O]_{\text{tot}}$). The repressor has two dimeric conformations, $R$ and $R^*$, which each have unique affinity constants and are linked by the thermodynamic equilibrium parameter $K_{RR^*}$.

Start by defining the following affinity constants in equilibrium:

$$K_{RR^*} = \frac{[R^*]}{[R]} \tag{2}$$

$$K_{RE} = \frac{[RE]}{[R][E]} \tag{3}$$

$$K_{R^*E} = \frac{[R^*E]}{[R][E]} \tag{4}$$

$$K_{RO} = \frac{[RO]}{[R][O]} \tag{5}$$

$$K_{R^*O} = \frac{[R^*O]}{[R^*][O]}. \tag{6}$$

We also need to define the total concentrations of operator, effector and repressor in terms of the individual bound and conformational states,

$$[O]_{\text{tot}} = [O] + [RO] + 2[REO] + [RE_2O] + [R^*O] + 2[R^*EO] + [R^*E_2O] \tag{7}$$

$$[E]_{\text{tot}} = [E] + 2[RE] + 2[RE_2] + 2[R^*E] + 2[R^*E_2]$$
$$+ 2[ROE] + 2[ROE_2] + 2[R^*OE] + 2[R^*OE_2] \tag{8}$$

$$[R]_{\text{tot}} = [R] + 2[RE] + [RE_2] + [R^*] + 2[R^*E] + [R^*E_2]$$
$$+ [RO] + 2[REO] + [RE_2O] + [R^*O] + 2[R^*EO] + [R^*E_2O]. \tag{9}$$

Of note are the various coefficients of 2. All of the singly bound effector species are degenerate since the effector can bind to either the left or right effector site, which gives rise to the statistical mass balancer 2. For Eq. (8), the doubly bound effector species have two effector molecules bound and hence are doubled.

The strategy was to write all of the equations in terms of the free species concentrations ($[R], [E], [O]$) and the equilibrium constants in Eqs. (2)–(6). Then we try to rearrange such that we can make polynomials of just $[E]$. The reasons will become apparent after we have done the above operations.

Starting with Eq. (9), we re-write using only free species and constants,

$$[RO] = K_{RO}[R][O] \tag{10}$$

$$[REO] = K_{RE}K_{RO}[R][E][O] \tag{11}$$

$$[RE_2O] = K_{RE}^2 K_{RO}[R][E]^2[O] \tag{12}$$

$$[R^*O] = K_{RR^*}K_{R^*O}[R][O] \tag{13}$$

$$[R^*EO] = K_{RR^*}K_{R^*E}K_{R^*O}[R][E][O] \tag{14}$$

$$[R^*E_2O] = K_{RR^*}K_{R^*E}^2 K_{R^*O}[R][E]^2[O] \tag{15}$$

$$[R]_{\text{tot}} = [R] + 2[R][E]K_{RE} + [R][E]^2 K_{RE}^2$$
$$+ [R]K_{RR^*} + 2[R][E]K_{RR^*}K_{R^*E} + [R][E]^2 K_{RR^*}K_{R^*E}^2$$
$$+ [R][O]K_{RO} + 2[R][O][E]K_{RO}K_{RE} + [R][O][E]^2 K_{RO}K_{RE}^2$$
$$+ [R][O]K_{RR^*}K_{R^*O} + 2[R][O][E]K_{RR^*}K_{R^*O}K_{R^*E} + [R][O][E]^2 K_{RR^*}K_{R^*O}K_{R^*E}^2. \tag{16}$$

We then make the following definitions,

$$\alpha_1 = 1 + K_{RR^*} \tag{17}$$

$$\beta_1 = 2K_{RE} + 2K_{RR^*}K_{R^*E} \tag{18}$$

$$\gamma_1 = K_{RE}^2 + K_{RR^*}K_{R^*E}^2 \tag{19}$$

$$\gamma_2 = 2K_{RO}K_{RE} \tag{20}$$

$$\delta_1 = K_{RO}K_{RE}^2 \tag{21}$$

$$\beta_2 = K_{RR^*}K_{R^*O} \tag{22}$$

$$\gamma_3 = 2K_{RR^*}K_{R^*O}K_{R^*E} \tag{23}$$

$$\delta_2 = K_{RR^*}K_{R^*O}K_{R^*E}^2. \tag{24}$$

Substituting into Eq. (16) and re-arranging to isolate $[R]$,

$$[R] = \frac{[R]_{tot}}{\alpha_1 + [E]\beta_1 + [E]^2\gamma_1 + [O](K_{RO} + [E]\gamma_2 + [E]^2\delta_1 + \beta_2 + [E]\gamma_3 + [E]^2\delta_2)}. \tag{25}$$

The equation has been organized such that polynomials in $[E]$ are apparent. As long as we only add and multiply polynomials, they can trivially be treated as symbolic functions for further simplification. We define the following polynomials,

$$B_1 = \alpha_1 + [E]\beta_1 + [E]^2\gamma_1 \tag{26}$$

$$B_2 = K_{RO} + \beta_2 + [E](\gamma_2 + \gamma_3) + [E]^2(\delta_1 + \delta_2). \tag{27}$$

Now substituting back into Eq. (25),

$$[R] = \frac{[R]_{tot}}{B_1 + [O]B_2}. \tag{28}$$

We next want to follow the same path for $[E]$ and $[O]$. Inspection of Eqs. (7)–(9) show that we have already done the most complicated case. We can then quickly arrive at,

$$[O] = \frac{[O]_{tot}}{1 + [R]B_2}. \tag{29}$$

The effector equation was similar, but it has a few extra coefficients of two within its equations. We define two more polynomials,

$$A_1 = \beta_1 + 2[E]\gamma_1 \tag{30}$$

$$A_2 = \gamma_2 + \gamma_3 + 2[E](\delta_1 + \delta_2). \tag{31}$$

Substituting into Eq. (8),

$$[E]_{tot} = [E] + [R][E]A_1 + [R][E][O]A_3. \tag{32}$$

We can then eliminate $[O]$ by substituting Eq. (29) into Eqs. (28) and (32). Since we can only multiply and add polynomials, we multiply the denominator of Eq. (29) on both sides. Substituting into Eq. (28),

$$[R]_{tot} + [R]B_2[R]_{tot} = [R]B_1 + [R]^2B_1B_2 + [R]B_2[O]_{tot}. \tag{33}$$

We then define the following polynomials,

$$\varphi_1 = B_1B_2 \tag{34}$$

$$\varphi_2 = B_1 + B_2([O]_{tot} - [R]_{tot}). \tag{35}$$

Substituting into Eq. (33),

$$[R]^2\varphi_1 + [R]\varphi_2 = [R]_{tot}. \tag{36}$$

The substitution of Eq. (29) into Eq. (32) requires the following definitions,

$$\psi_1 = [E]A_1B_2 \tag{37}$$

$$\psi_2 = [E](B_2 + A_1 + A_2[O]_{\text{tot}}) - B_2[E]_{\text{tot}}. \tag{38}$$

We then arrive at,

$$[R]^2\psi_1 + [R]\psi_2 = [E]_{\text{tot}} - [E]. \tag{39}$$

We now have two equations (Eqs. (36) and (39)) with two unknowns ($[R]$ and $[E]$). In principal we can get this down to a single equation, but in order to do so the final polynomial becomes of a much higher order which prevents accurate computational solutions.

The strategy was then to guess at the free effector concentration to calculate Eqs. (34), (35), (37) and (38). Equations (36) and (39) can then be solved for $[R]$ by looking for the roots to the equation. When the correct free effector concentration ($[E]$) is found the roots of Eqs. (36) and (39) will converge. By minimizing the difference between the roots a solution can be reached. All other concentrations are then trivial to calculate once $[R]$ and $[E]$ are known. Custom Matlab (Mathworks) software was written to numerically solve the MWC equilibria (Matlab File Exchange ID #40602).

The accuracy of the solution was easily checked by using the bound and free species concentrations to calculate the total species concentrations and thermodynamic parameters. Calculated values should agree with input values.

Five independent thermodynamic parameters ($K_{RE}$, $K_{R*E}$, $K_{RO}$, $K_{R*O}$, and $K_{RR*}$) were used for each model and all data points were simultaneously fit using a standard non-linear least squares algorithm in Matlab.

A Monte Carlo approach was used to estimate error in the fit parameters. The known error of the experiment was used to generate data sets with random error. 100 such data sets were generated and a non-linear least squares fitting algorithm was used to fit the thermodynamic parameters. Standard deviation of these fit thermodynamic parameters was used as the error of the best fit for the actual data set.

## RESULTS AND DISCUSSION

### Measuring the *in vivo* concentration of the lac repressor

We sought a method where we could simultaneously measure lac repressor concentration and transcriptional regulation and thus chose to fluorescently tag the repressor. The fluorescent protein mCherry was chosen due to minimal auto-fluorescence from MOPS minimal media and minimal spectral overlap with our reporter gene YFP. Furthermore, a dimeric Lac-mCherry fusion construct was known to be functional *in vivo* (*Lau et al., 2004*). Assembly of the dimer is necessary for operator DNA binding and addition of C-terminal additions increases affinity for DNA (*Chen, Alberti & Matthews, 1994*; *Tungtur et al., 2011*). We measured the ability of the Lac-mCherry fusion construct to form dimers by native gel electrophoresis and found that >95% of the purified protein was dimeric

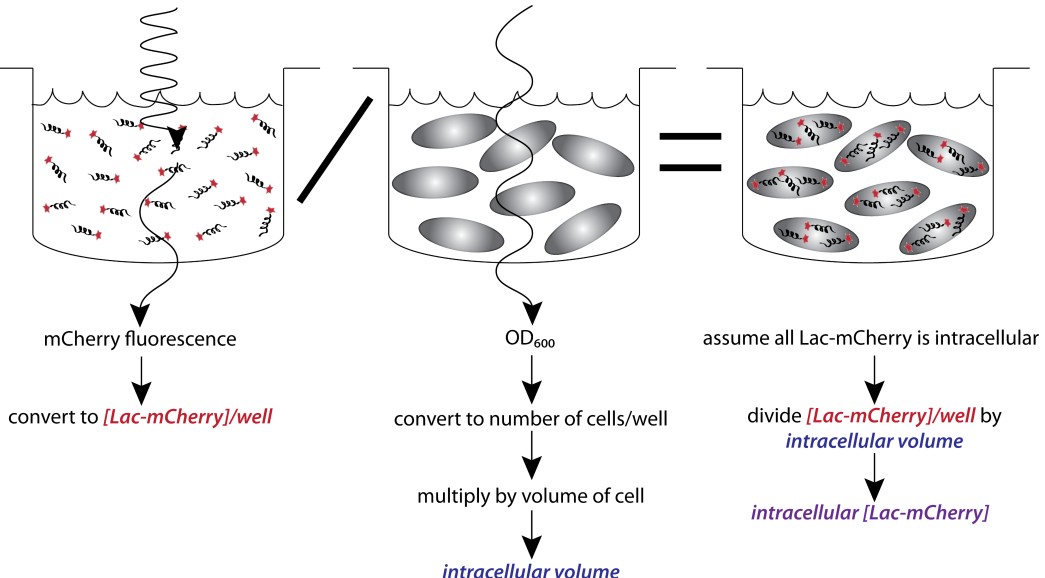

**Figure 2** **Measuring intracellular Lac-mCherry concentration.** Raw mCherry fluorescence and $OD_{600}$ are measured on a plate reader. Calibration curves for both were established given our experimental setup (cell line, plasmids, media, amount of sample loaded, plates and plate reader). Raw fluorescent signal was converted to concentration of Lac-mCherry per well. Raw $OD_{600}$ signal was converted to the fraction of well volume that was intracellular. Dividing Lac-mCherry well concentration by intracellular volume fraction effectively concentrates the Lac-mCherry to be intracellular. These two measurements, combined with the appropriate calibrations, allow a quick and accurate measurement of intracellular Lac-mCherry concentration.

(Fig. S1). The goal was to measure raw mCherry fluorescence and $OD_{600}$ in growing *E. coli* cells and convert those measurements to an intracellular concentration of lac repressor (Fig. 2).

A linear relationship was established for $OD_{600}$ and cell count. We estimated the volume of *E. coli* growing in glucose supplemented minimal media to be $1 \times 10^{-15}$ L (*Kubitschek & Friske, 1986*). We then measured $OD_{600}$, calculated the number of cells and multiplied by volume of the cell to calculate the fraction of the well that was intracellular. A linear relationship was also established for purified Lac-mCherry fluorescence and concentration of Lac-mCherry (Fig. S2).

We assume all of the Lac-mCherry was intracellular; therefore we divided the Lac-mCherry concentration by the fraction of volume that was intracellular. Using this method, we quickly and accurately measured *in vivo* Lac-mCherry concentrations.

## Intracellular Lac-mCherry concentration in EPB229 cells

EPB229 cells expressing Lac-mCherry and the reporter plasmid were grown in varying concentrations of the inducer IPTG. Intracellular concentration of Lac-mCherry was calculated from mCherry fluorescence and $OD_{600}$ and found to be $664 \pm 90$ nM at mid-log growth phase ($OD_{600} = 0.6$). As expected for a constitutively expressed gene, minimal variation was seen with IPTG and cell growth (Fig. 3A).

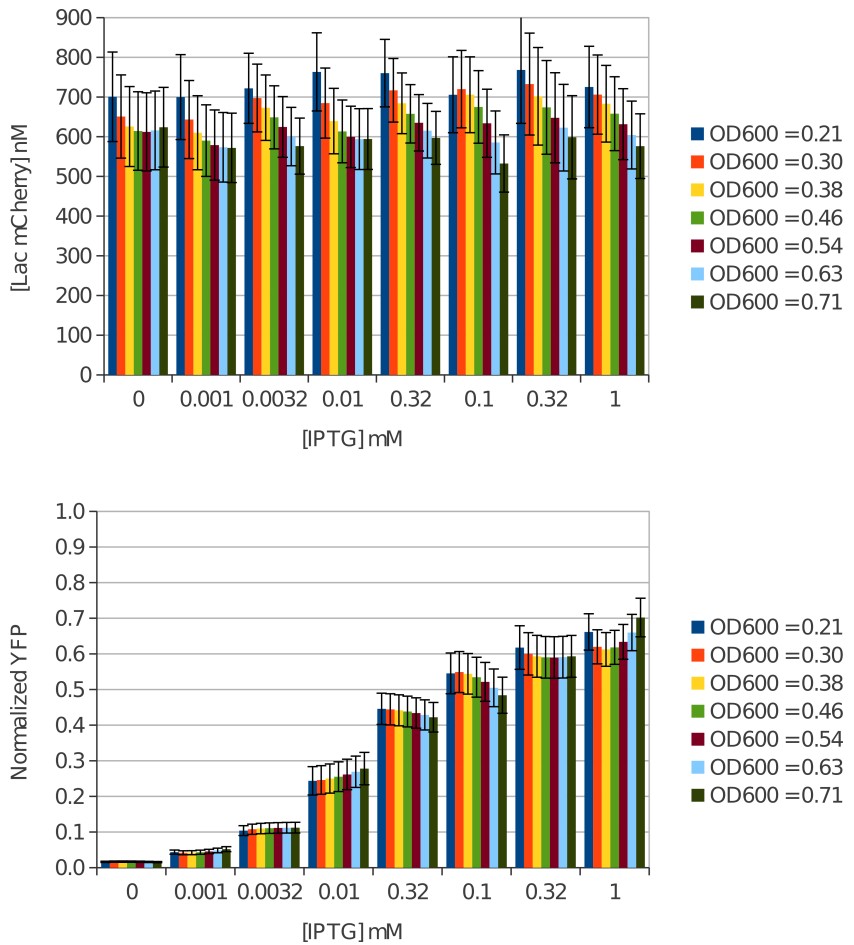

**Figure 3** *In vivo* **Lac-mCherry and YFP regulation show no growth dependence.** (A) Lac-mCherry was calculated for growing *E. coli* cells and found to have minimal $OD_{600}$ dependence. As expected for a constitutively expressed gene, there was no change in Lac-mCherry concentration with increasing IPTG concentration. (B) Normalized YFP was simultaneously measured and again no $OD_{600}$ dependence was found throughout the exponential growth phase. In stark contrast to the Lac-mCherry concentration, a distinct induction profile was measured for YFP as a function of IPTG.

We then converted to molecules per cell,

$$6.6 \times 10^{-7} \, \text{M} * 1 \times 10^{-15} \, \frac{\text{L}}{\text{cell}} * 6.022 \times 10^{23} \, \frac{\text{molecules}}{\text{mole}} = 397 \, \frac{\text{molecules}}{\text{cell}}. \tag{40}$$

We have previously estimated the copy number of our plasmid to be ∼10–20 plasmids/cell (*Daber, Sharp & Lewis, 2009*). This corresponds to approximately 20–40 Lac-mCherry dimers per plasmid which agrees well with previous estimates of ∼40 Lac repressor dimers per plasmid for our promoter (*Oehler et al., 1994*).

## Measuring the *in vivo* regulation of YFP

In addition to mCherry fluorescence and $OD_{600}$ measurements, YFP fluorescence was measured in cells as a function of IPTG. Unregulated expression was established by measuring $OD_{600}$ and YFP in cells co-transformed with O1 YFP reporter and a plasmid

Peer J

which does not contain any repressor (pABD34). These positive control cells were grown in tandem with cells containing both reporter and repressor and grown in a variety of IPTG concentrations.

Positive controls showed no IPTG dependence as expected, so data from every sample was combined to determine an overall positive control polynomial fit. YFP fluorescence was seen to increase as cells grow as would be expected due to the increased number of cells per μL. We remove this bias and normalize regulated YFP expression by dividing by the positive control fit curve.

Normalized YFP expression was then measured as a function of $OD_{600}$ and IPTG (Fig. 3B). Almost no $OD_{600}$ dependence can be noted in the induction profile. The YFP signal was repressed without IPTG and was approximately $1.7 \pm 0.2\%$ of unregulated expression. Upon induction with saturating IPTG we saw a robust YFP increase to approximately $61 \pm 5\%$ of the unregulated expression.

## Measuring the *in vitro* regulation of mRNA

While the *in vivo* experiment measured translation product (fluorescing YFP) we know the lac repressor actually regulates mRNA production. Previously, our lab has determined a linear relationship between mRNA and fluorescence protein signal allowing us to use fluorescence as a proxy for mRNA regulation *in vivo* (*Daber & Lewis, 2009*). The situation *in vitro* was reversed; it was much easier to measure mRNA production.

We sought to measure *in vitro* regulation of mRNA using a setup that mimics the *in vivo* YFP regulation as close as possible. To this end we purified the Lac-mCherry construct used *in vivo* with the addition of a C-terminal 6xHis tag for purification. We also used a 450 base pair linearized DNA strand with a T7 promoter controlled by the O1 operator in the same location as the O1 operator relative to the natural lac Z/Y/A promoter.

We used the Maxiscript T7 *in vitro* transcription kit (Ambion) which produces mRNA from linearized DNA with a T7 promoter. We then measured incorporation of radioactive labeled CTP into mRNA. The T7 promoter was modified to add an O1 operator DNA site and we were able to modulate Lac-mCherry and IPTG concentrations. A positive control of constitutive mRNA production was established by not adding any Lac-mCherry.

We first established that radioactively labeled mRNA was linearly observable by constitutively producing mRNA and loading various dilutions onto polyacrylamide gels and established that mRNA concentration was linearly related to the concentration of mRNA loaded on the gel. Positive controls were included for every experiment and were used for normalization.

The additional benefit of *in vitro* transcription was the flexibility in dosing not only IPTG, but also Lac-mCherry. We therefore first titrated Lac-mCherry with and without IPTG present (1 mM) (Fig. 4A). As expected, increasing concentration of Lac-mCherry decreased mRNA production. Furthermore, addition of IPTG returns mRNA signal to near constitutive levels.

We then titrated IPTG at a fixed Lac-mCherry concentration (Fig. 4B). The induction of mRNA was seen to very closely resemble that of the *in vivo* data, but it was noticeably

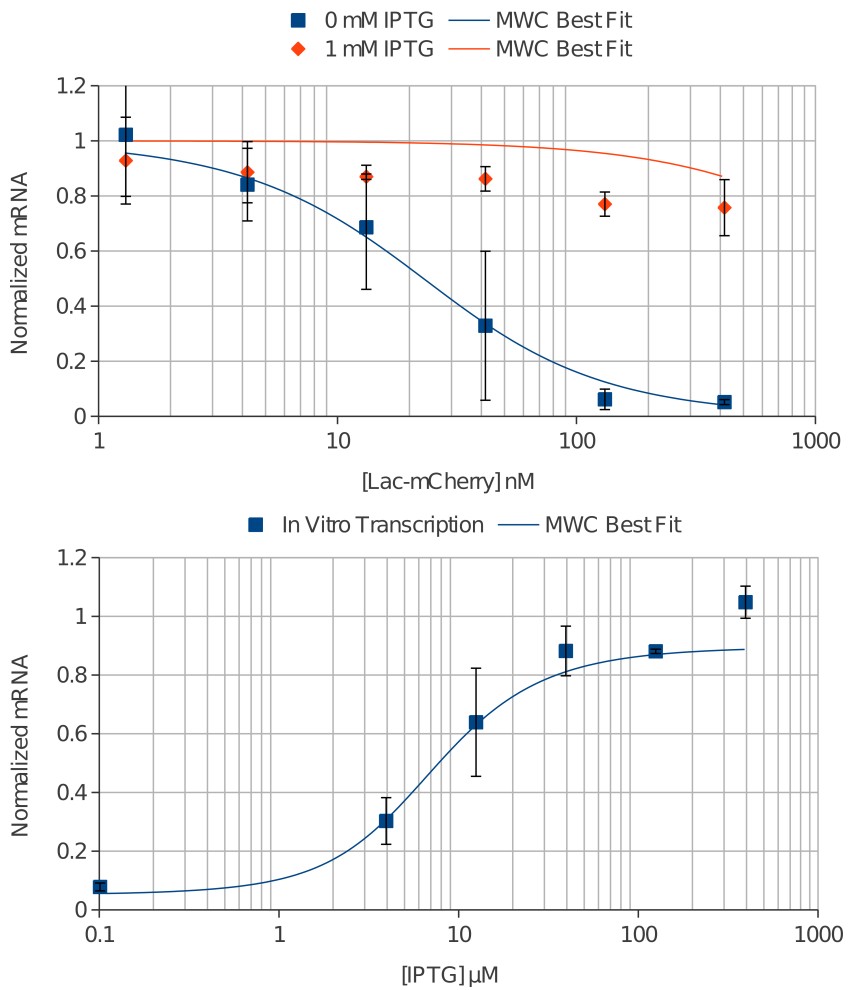

**Figure 4** *In vitro* **transcription controlled by the lac repressor was accurately fit by the MWC model.** (A) Lac-mCherry was added at varying concentrations with 11 nM O1 DNA and mRNA was quantified (blue squares). The repression was relieved upon addition of 1 mM IPTG (orange diamonds). The data was globally fit by the MWC model and an accurate solution was found for the Lac-mCherry titration (solid blue line). The model predicts higher induction than was measured experimentally (solid orange line). (B) IPTG was added at varying concentrations with 333 nM Lac-mCherry and 11 nM O1 DNA and again mRNA was quantified (blue squares). A robust induction profile was measured showing induction up to approximately 80% of constitutive expression. The global fit also accurately fits the IPTG titration data (solid blue line).

leakier. Maximal repression was about $7.8 \pm 1.3\%$ and maximal induction was approximately $88 \pm 9\%$.

## Modeling using MWC thermodynamic equilibrium

Finally, we sought to model both the *in vivo* and *in vitro* data using the Monod, Wyman, and Changeux (MWC) model of thermodynamic equilibrium. Previously, we have relied upon approximate solutions of the lac genetic switch equilibrium to model *in vivo* induction profiles. This solution assumes that the total repressor concentration greatly exceeds operator concentration ($[R]_{\text{tot}} \gg [O]_{\text{tot}}$). This condition does not hold for our

*in vitro* experiment where we titrated in Lac-mCherry nor would it necessarily be true in all *in vivo* systems. Therefore, we sought a solution to the equilibrium that held for every potential input. An assumption free solution to the MWC model was found and was solved in detail in the methods.

**Using the assumption-free solution to measure thermodynamic parameters**

Experimentally we know the total concentrations ($[R]_{tot}$, $[E]_{tot}$, $[O]_{tot}$) and normalized transcription/expression ($[O]/[O]_{tot}$). We want to measure the thermodynamic constants ($K_{RR*}, K_{RE}, K_{R*E}, K_{RO}, K_{R*O}$). This leaves 5 independent constants in the MWC model to fit to the experimental data. The large number of independent constants results in a myriad of non-unique solutions to the equations. This complication was limited by the following algorithm.

First, since it was widely reported to be effectively zero, $K_{R*O}$ was set to be very, very small ($1 \times 10^{-10}$ nM$^{-1}$). This leaves four independent parameters.

Next, it had been observed from previous studies that the ratio of $K_{R*E}$ to $K_{RE}$ was well defined when the concentration of repressor greatly exceeds that of operator. Under this assumption, a simpler solution of the MWC equilibrium exists as previously reported (*Daber, Sharp & Lewis, 2009*). We isolated a subset of the *in vitro* data where this condition was true and used a non-linear least squares fitting algorithm to measure the ratio $X = K_{R*E}/K_{RE}$ as a function of conformational equilibrium. The ratio was seen to asymptote at approximately 13.75. This value was then used to reduce the number of independent constants to 3 ($K_{RR*}, K_{RE}$, and $K_{RO}$).

We then simultaneously fit the *in vitro* data to obtain the best fit thermodynamic parameters using a non-linear least squares algorithm in Matlab (Table 1). The model accurately fit both the lac repressor (Fig. 4A) and IPTG doping (Fig. 4B) *in vitro* transcription experiments. The fit values agree well with values obtained in the literature with the exception of repressor-DNA affinity. The repressor-DNA affinity ($K_{RO}$) was measured to be $0.4 \pm 0.2$ nM$^{-1}$. This was significantly weaker than the 100–3333 nM$^{-1}$ that has been measured previously (*Sharp, 2011*). It does agree well with an estimated value of 1 nM$^{-1}$ for lac repressor-DNA affinity that prevails under conditions within the *E. coli* cell (*Müller-Hill, 1996*). All measurements from the literature were for tetrameric lac repressor bound to a single O1 operator. *Chen, Alberti & Matthews (1994)* showed that dimeric lac repressor (with the 11 C-terminal residues deleted) had 100 fold weaker affinity to operator DNA but the addition of a C-terminal linker restored dimeric affinity to that of tetramer. This would indicate the only difference between dimeric lac repressor and tetrameric lac repressor affinity to a single operator DNA is due to its ability to dimerize, which is consistent with the "dimer of dimers" viewpoint of the full tetrameric lac repressor; tetramerization enhances dimerization and can be replaced by C-terminal additions. The Lac-mCherry construct is nearly all in the dimeric state (Fig. S1) allowing for direct comparison with these tetrameric literature values. The dimeric nature of the Lac-mCherry construct does not explain the weaker affinity for operator DNA.

**Table 1  Fit values from the MWC models compared with literature values.** All fit parameters agree with the exception of repressor–operator DNA affinity (KRO).

| | This study | Daber, Sharp and Lewis[a] | Daber, Sochor and Lewis[b] | Sharp, Set 1[c] | Sharp Set 2[c] | Sharp, Set 3[c] | Müller-Hill[d] |
|---|---|---|---|---|---|---|---|
| $K_{RR*} = [R^*]/[R]$ | $6.3 \pm 3.4$ | $2 \pm 0.5$ | $5.8 \pm 0.07$ | | | | |
| $K_{RO}$ (nM$^{-1}$) | $0.42 \pm 0.21$ | | | 3330 | 100 | 1510 | 1 |
| $K_{RE}$ (nM$^{-1}$) | $5.6 \times 10^{-5} \pm 1.8 \times 10^{-5}$ | | $6 \times 10^{-5} \pm 2 \times 10^{-7}$ | | | | |
| $K_{R*E}$ (nM$^{-1}$) | $7.6 \times 10^{-4} \pm 2.5 \times 10^{-4}$ | | $5 \times 10^{-4} \pm 5 \times 10^{-6}$ | $2.3 \times 10^{-4}$ | $2.3 \times 10^{-4}$ | $2.3 \times 10^{-4}$ | |
| $K_{R*O}$ (nM$^{-1}$) | $1.0 \times 10^{-10}$ | | | | | | |
| $R_{tot}$ (nM) [with 40% crowding] | $664 \pm 90 \; [1660 + 225]$ | | | | | | |
| $r = K_{RO}^* R_{tot}$ [with 40% crowding] | $278 \; [697]$ | $150 \pm 50$ | $150 \pm 50$ | | | | |
| $X = K_{R*E}/K_{RE}$ | $13.7 \pm 0.13$ | $15 \pm 3$ | $8.28$ | | | | |

Notes.

[a] *Daber, Sharp & Lewis (2009)*.

[b] *Daber, Sochor & Lewis (2011)*.

[c] *Sharp (2011)*.

[d] *Müller-Hill (1996)*.

The thermodynamic equilibrium value ($6.3 \pm 3.3$) does not significantly differ from that measured previously by our group. The repressor-IPTG affinity ($7.6 \times 10^{-4} \pm 2.5 \times 10^{-4}$ nM$^{-1}$ for the higher affinity conformation) was found to be slightly higher than previously published values ($2.3 \times 10^{-4}$ nM$^{-1}$) but it was generally within agreement. The ratio of affinities for the two conformations (13.7) was in good agreement with previously measured values.

## Using the *in vitro* thermodynamic parameters to predict *in vivo* genetic regulation

The raison d'être for *in vitro* measurements was to inform what was occurring *in vivo*. One of the central difficulties in using *in vitro* measurements was the lack of a well enough defined *in vivo* system to directly compare it with. Furthermore, a model was required which can accurately function in both circumstances and provide useful predictions. We then sought to fully define our *in vivo* experiment to model it with the *in vitro* determined thermodynamic parameters.

We estimated the copy number of our operator reporter plasmid to be $\sim$20 copies per cell (*Daber, Sharp & Lewis, 2009*). This then gives,

$$[O]_{tot} = \frac{20 \text{ molecules}}{6.02 \times 10^{23} \frac{\text{molecules}}{\text{mole}}} * \frac{1}{1 \times 10^{-15} \text{ L}} * 1 \times 10^9 \frac{\text{nM}}{\text{M}} = 33 \text{ nM}. \tag{41}$$

The strain of *E. coli* used has the lac genetic switch deleted from the genome; therefore lac permease was also deleted. It was then assumed that IPTG enters the cell through passive diffusion and has the same concentration as the media.

Figure 5A shows the simulated *in vivo* data (solid blue line) along with experimentally determined values (blue squares). The model predicts both higher leakiness (2.7%

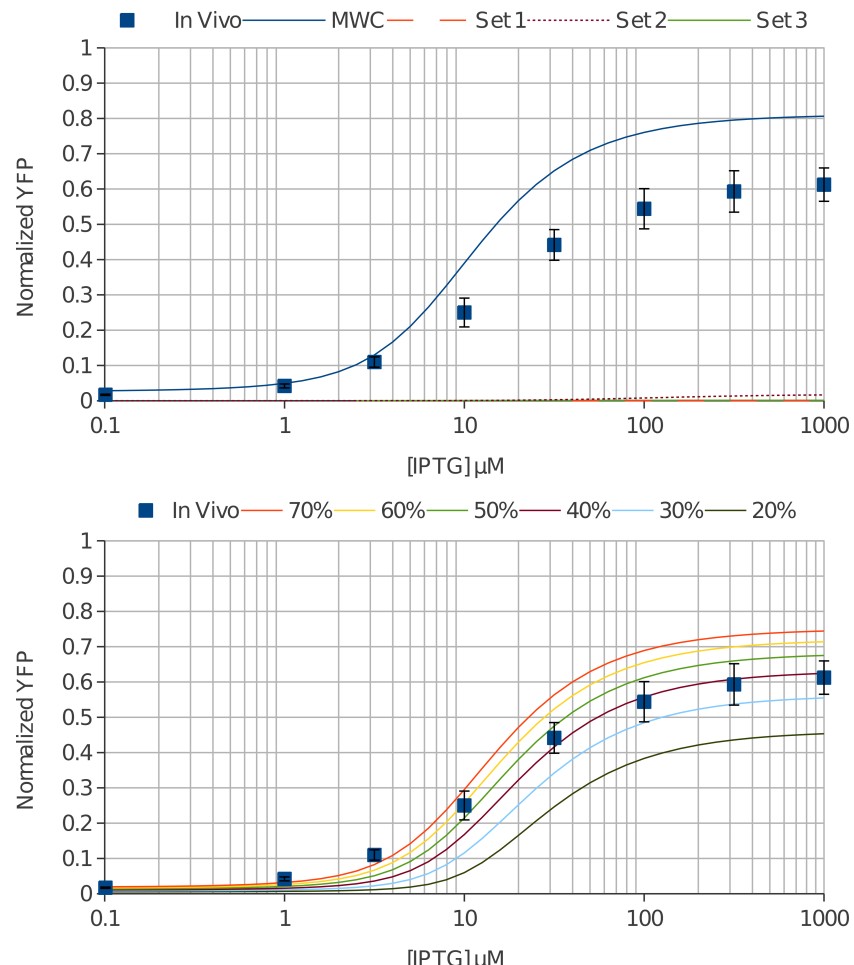

**Figure 5** ***In vivo* regulation by the lac repressor was accurately predicted with molecular crowding.** (A) YFP under control of the lac repressor was measured in *E. coli* cells at varying concentrations of IPTG (blue squares). We used the measured intracellular concentration of the lac repressor (660 nM) and the fit values from *in vitro* transcription to predict the *in vivo* induction curve with the MWC model (solid blue line). The model predicts more YFP signal at all concentrations of IPTG. Our repressor-DNA affinity was much lower than previously published values, so we also modeled three curated data sets (*Sharp, 2011*) (dashed orange, dotted purple, and solid green lines). All three predict greatly over-repressed YFP expression and do not fit the *in vivo* data. (B) Molecular crowding was known to play a significant role in cells. We modeled this by estimating the available volume in percentage for our repressor and calculated an effective repressor concentration. We modeled several percentages and 40–60% available volume (solid purple, green and yellow lines) accurately reproduces the *in vivo* regulation from the *in vitro* transcription derived thermodynamic constants. 40% crowding corresponds to an effective repressor concentration of 1.6 μM.

predicted versus $1.7 \pm 0.2\%$ observed) and higher maximal induction (80% predicted versus $61 \pm 5\%$ observed) than was measured *in vivo*. This indicates that there are additional effects not being accounted for in the *in vitro* data. It has been postulated that non-specific DNA binding of repressors could play a significant role (*Tungtur et al., 2011*), however this should have the effect of decreasing the effective lac repressor concentration since the non-specific DNA will competetively bind with operator DNA for lac repressor.

We see the opposite in our data; the lac repressor concentration appears higher *in vivo* than we are measuring.

There is a known molecular crowding effect in living cells due to the density of molecules which will increase the *effective* concentration of molecules. We can quickly model the effect of crowding by decreasing the available space for the lac repressor and estimating its effective concentration,

$$[R]_{tot}^{eff} = \frac{[R]_{tot}}{\% \text{ available space}}. \tag{42}$$

Figure 5B shows the effect of including molecular crowding on the predicted *in vivo* induction curve. The model shows excellent agreement with experiment at a molecular crowding of 40%–60% which estimates effective *in vivo* repressor concentration to be 1.1–1.6 µM (Leakiness: $1.3 \pm 0.3\%$ predicted versus $1.7 \pm 0.2\%$ observed; Maximal expression: $67 \pm 4\%$ predicted versus $61 \pm 5\%$ observed). Furthermore, this value agrees well with estimates of 20%–40% available space *in vivo* (*Kubitschek & Friske, 1986*).

Since there was a notable deviation in repressor-DNA affinity with previous *in vitro* measurements, the same analysis was carried out for the three curated data sets from *Sharp (2011)*. Using the values from the literature, we find that they do not in any case come close to replicating our *in vivo* data (Fig. 5A, orange dashed, purple dotted line, and solid green lines). The DNA affinities are much too high for the measured DNA and repressor concentrations. At these affinities the switch was essentially completely off and cannot be induced with any concentration of IPTG. Crowding only enhances the deviation from experiment as it further increases the concentration of repressor.

## Simulating native *in vivo* lac genetic switch phenotype

The thermodynamic constants from our *in vitro* data better represents our *in vivo* model system. The question then is: which set of thermodynamic parameters could effectively regulate the native lac genetic switch?

Essentially we have rebuilt the lac operon with the *lacZ*, *lacY* and *lacA* polycistronic message replaced by the reporter gene YFP and the dimeric lac repressor constitutively expressed by its native promoter. We have a higher copy number of both the reporter and repressor plasmids (~20 copies per cell) which increased both the operator and repressor concentrations above that normally found in the cell. A secondary deviation was the removal of the tetramerization domain and multiple operator DNA sites (O2 and O3 additionally exist on the genome) which simplified our analysis. The cooperativity of the native tetrameric lac repressor is known to decrease leakiness approximately 10-fold, so we expect a dimeric lac repressor with one operator (O1) to have some leakiness in its repression (*Oehler et al., 1994*).

As previously mentioned, *in vivo* lac repressor dimer concentration was measured to be ~40 dimers per cell, which gives,

$$[R]_{tot} = \frac{40 \text{ molecules}}{6.02 \times 10^{23} \frac{\text{molecules}}{\text{mole}}} * \frac{1}{1 \times 10^{-15} \text{ L}} * 1 \times 10^9 \frac{\text{nM}}{\text{M}} = 66 \text{ nM}. \tag{43}$$

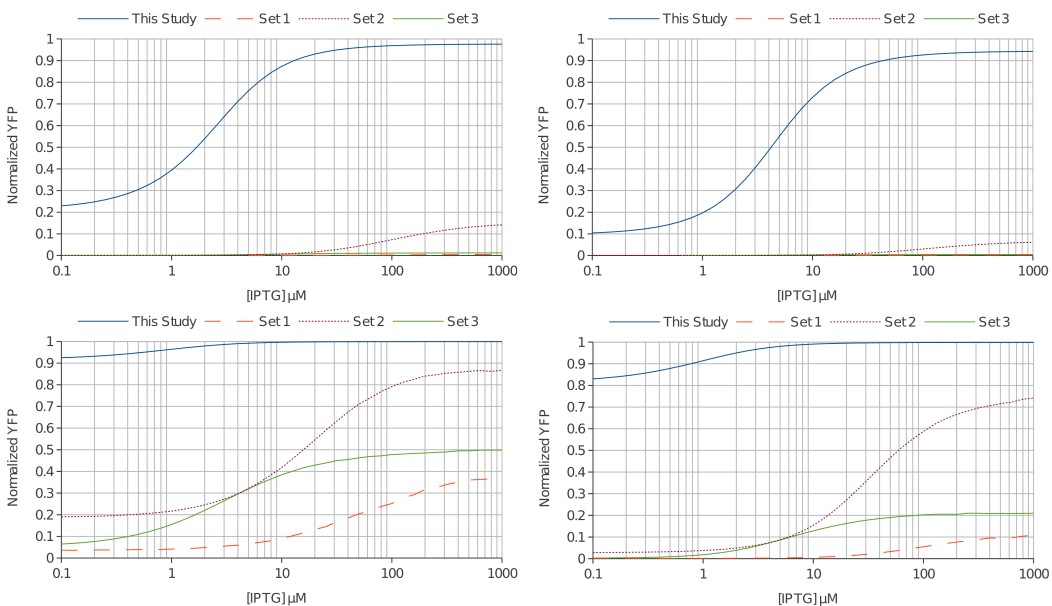

**Figure 6 Simulating a simplified lac operon from *in vitro* derived thermodynamic constants.** The correct repressor-DNA affinities must be able to provide robust switching under conditions naturally experienced by *E. coli*. With this in mind, we modeled a dimeric lac repressor regulating a gene with a single operator sequence. (A) The natural lac promoter makes ∼66 nM of lac repressor dimer and one operator was at ∼1.7 nM in the cell. We modeled these conditions for the thermodynamic parameters from this study and for the three curated data sets of Sharp. The predicted curve from this study shows a reasonable repression and induction profile (solid blue line). Only Set 2 from Sharp was weakly inducible (dotted purple line). (B) Including molecular crowding (40% available volume) enhances the situation. The curated data sets do not make useful switches. Alternately, the predicted induction curve from *in vitro* transcription derived constants shows a leaky switch that induces very well (solid blue line). (C) We next sought to model the minimal possible repressor to find a condition where the curated data sets produce reasonable induction curves. 1 molecule of dimer per cell (∼1.7 nM) does show good induction profiles for set 2 (dotted purple line) and set 3 (solid green line). Set 1 still shows a switch that can marginally be induced and would likely not be useful (dashed orange line). (D) Molecular crowding effects again enhance the repressor concentration and only set 2 could reasonably regulate a gene (dotted purple line). The values from this study (solid blue line) predict a very leaky switch. Although the second curated set could effectively regulate the gene at this concentration, in reality a single dimer and single operator DNA binding would be dominated by stochastic events creating an inherently unstable switch.

And we know there is one operator per cell in the native system,

$$[O]_{\text{tot}} = \frac{1 \text{ molecules}}{6.02 \times 10^{23} \frac{\text{molecules}}{\text{mole}}} * \frac{1}{1 \times 10^{-15} \text{ L}} * 1 \times 10^9 \frac{\text{nM}}{\text{M}} = 1.7 \text{ nM}. \tag{44}$$

Using these values, along with the experimentally determined binding constants derived from this study and those curated by Sharp, we simulated dimeric lac repressor induction curves at native conditions. Figure 6A shows that the values determined in this study predict a leaky repressor that was maximally inducible. The much higher DNA affinities of the curated data sets all produce over-repressed curves that do not show good induction.

The over-repression was even more prominent as cell crowding was considered. Using the value of 40%, which gives $R_{\text{tot}} = 66 \text{ nM}/0.4 = 165 \text{ nM}$, we find that the over-repression

of the high affinity DNA sets all produce curves that weakly induce or do not induce at all (Fig. 6B). The predicted curve using our thermodynamic parameters again provides reasonable induction (~10% leakiness up to ~95% maximal induction). While this level of leakiness would be intolerably high for efficient regulation of the lac operon, the restoration of the tetramerization domain would significantly decrease the leakiness while minimally impairing inducibility.

If we consider the lowest possible concentration of lac repressor (1 molecule/cell; $R_{tot} = 1.7$ nM; with 40% crowding $R_{tot} = 4.25$ nM) we find that second curated data set does produce reasonable induction curves, even if 40% crowding was taken into consideration (Figs. 6C and 6D). Unfortunately, in this regime the binding would be highly stochastic and hence noisy, which would not produce stable repression. Furthermore, this level of repressor expression does not agree with published values. While it was technically possible for these affinities to be accurate, it was highly improbable. The first and third data sets would require less than 1 molecule of dimeric lac repressor per cell to be functionally useful according to our model.

Given the wide range of repressor–operator DNA affinities (100 nM–3333 nM) it can be reasonably concluded that these values must contain significant artifacts from the experimental techniques. Techniques such as gel shift assays, where molecular "caging" effects are known to be significant, and nitrocellulose filter binding assays, where the binding was removed from the solution phase, were used to create the curated data sets. Our measurement of repressor-DNA binding affinity did require an indirect measurement, namely transcription, but it did occur in the solution phase. We attribute the difference in values to differences in experimental setup.

## CONCLUSIONS

We have reproduced the transcriptional regulation of the lac repressor dimer *in vitro* and shown that it accurately reproduces the *in vivo* repression of YFP under control of the lac repressor. Accurate modeling of the *in vivo* data required an estimate of 40%–60% cellular crowding in the cell, which agrees with previous estimates. Non-specific DNA binding and IPTG uptake did not appear to have any significant effect. Crowding could be tested *in vitro* through crowding agents such as bovine serum albumin (BSA) or polyethylene glycol (PEG) (*Ellis, 2001*). Alternative explanations are potentially possible such as fluctuations in the size of the *E. coli*. What was essentially important was that the concentration of lac repressor in the cell greatly affects the maximal induction given our thermodynamic parameters. The curve was extremely sensitive in that region to changes in repressor concentration. So only an approximately two-fold increase in repressor concentration was sufficient to replicate the *in vivo* data. Whether the lac repressor concentration was increased by molecular crowding or by decreased *E. coli* volume would have to be tested by further experiments.

The measured thermodynamic binding parameters match well for IPTG binding and conformational equilibrium, except there was significantly lower repressor/operator DNA affinity measured (by approximately 3–4 orders of magnitude). This discrepancy was

modeled and it was demonstrated that the affinity measured in this study was capable of reproducing not only the *in vivo* data from this study, but also can predict reasonable induction curves at concentrations of repressor and DNA that are naturally seen by *E. coli*. We therefore conclude that lac repressor DNA affinity was significantly weaker than previous *in vitro* measures and more in line with the estimates for repressor-DNA affinity at *in vivo* conditions where we do find good agreement with previously published values.

A limitation to this study exists regarding the dimeric lac repressor construct fused to mCherry. The monomer–monomer assembly to dimer is not considered in our current MWC model as it is assumed all of the repressor is in the dimeric state. The addition of the C-terminal mCherry appears to promote dimerization (Fig. S1) and therefore this study ignores this complication as only dimeric repressor is visualized in the native gel. A more complete thermodynamic model, including dimerization or tetramerization for full length lac repressor, would be a significant improvement upon our work.

Finally, this study highlights the difficulty in using *in vitro* data generated from experimental techniques that are divorced from conditions closer to that of the cell. Experimental artifacts may greatly overshadow actual values, which should come as no surprise in the case of lac repressor binding to operator DNA where the published binding constant has changed 33-fold as experimental techniques have changed. The difficulty in *in vitro* measurements was well known in the field as was evidenced by the large consideration given to differences in buffer conditions (*Ha et al., 1992*), DNA length (*Khoury et al., 1990*), and even hydrostatic pressure (*Royer, Chakerian & Matthews, 1990*). Techniques such as gel filtration or nitrocellulose filter binding assays are excellent at differentiating binding strength between point mutants; they are limited in comparison with *in vivo* results. Using experimental setups which more closely mimic the *in vivo* system can significantly improve the ability of the predictive capabilities of *in vitro* experiments. They do come with the caveat that the data interpretation was not as straightforward as simple binding experiments.

## ACKNOWLEDGEMENTS

I would like to thank Dr. Mark Goulian for the kind gift of the YFP and mCherry genes along with the EPB229 cell line. I would like to thank Dr. Kristin Lynch, Sandya Ajith, Nicole Martinez, Chris Yarosh and Michael Mallory for guidance and assistance performing the *in vitro* transcription experiments. I would also like to thank Dr. Elizabeth Sweeny for assistance in making figures.

### Funding

This was funded by the National Institutes of Health Structural Biology Training Grant: 5-T32-GM-008275-24. The funders had no role in study design, data collection and analysis, decision to publish, or preparation of the manuscript.

## Grant Disclosures
The following grant information was disclosed by the author:
National Institutes of Health: 5-T32-GM-008275-24.

## Competing Interests
The authors declare there are no competing interests.

## Author Contributions
- Matthew Almond Sochor conceived and designed the experiments, performed the experiments, analyzed the data, contributed reagents/materials/analysis tools, wrote the paper, prepared figures and/or tables, reviewed drafts of the paper.

## Supplemental Information
Supplemental information for this article can be found online at http://dx.doi.org/10.7717/peerj.498.

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
