# Peer review of "In vitro transcription accurately predicts lac repressor phenotype in vivo in Escherichia coli"

_PeerJ, doi:10.7717/peerj.498_

## Round 0.1 · original submission · Major Revisions

Please revise the manuscript according to the reviewers' reports. The revised manuscript will need to be re-reviewed.

Reviewer 1 ·

Basic reporting

This article is written in a manner that suggests that it is derived from a report or perhaps a thesis draft. Substantial editing for accuracy of statements and clarity would both improve and clarify the presentation. Careful proofreading and better phrasing would improve the written manuscript.

Experimental design

The experiments presented are potentially interesting as a mechanism for examining the concentration of lac repressor in the cell and comparing in vivo and in vitro impacts of this regulatory protein on transcription. However, no evidence is provided that the fluorescence of the C-terminal m-Cherry tag is directly proportional to the repressor concentration. Further, there is a substantial problem in that the lac repressor dimer does NOT bind to operator with the affinity of the wild-type lac repressor tetramer. Assembly from monomer to dimer is required for binding, and the monomer-monomer affinity is significantly less than the affinity of the assembled dimer for operator DNA unless additions at the C-terminus are added to promote assembly [e.g., see Chen and Matthews (1994) J. Biol. Chem. 33, 8728; Tungtur et al. (2011) Biophys. Chem. 159, 142]. The C-terminal addition of the fluorescent protein (given that this family of proteins has been used to assess interactions of fused proteins) or the C-terminal His-tag (given its likely positive charge) would be unlikely to promote higher affinity for assembly to dimer and might even interfere with this association, further complicating interpretation of the data presented.

Validity of the findings

Without in vitro operator DNA binding data on the dimeric repressor with the attached fluorescent protein and His-tag to demonstrate identity of operator binding affinity with that for wild-type tetrameric lac repressor, the experiments presented are impossible to interpret quantitatively in terms of the modified repressor binding to DNA. Linking the binding constants for monomer-monomer assembly to operator binding is required for the mathematical modeling of this system. The lower affinity of the dimeric protein for operator DNA deduced by the author likely derives directly from thermodynamic linkage of monomer-monomer assembly (to dimer) with dimer binding to operator DNA. This important linkage would need to be addressed both in the experiments and in the mathematical modeling before further consideration of this manuscript. The conclusions as they stand are not supported by the data.

·

Basic reporting

In general, the manuscript was very clearly written. A few minor points:

The verb tense occasionally switches between past and present tense. Please standardize. (The most common offense is "is" instead of "was".)

On page 12 line 268 – the paragraph has redundant sentences.

“Quantitate” should be “quantify”.

R* is not defined in the text but is in a figure legend. It was confusing in the current layout (with tables/figures separated from the text). It might be more clear in the final version but it would be good to define it in the text.

Various section of the manuscript report the in vitro and in vivo works in different orders. It would help the reader to report them in a standard order.

Experimental design

Buffer conditions were not noted anywhere in the paper for in vitro measurements. As discussed below, these are very important in interpreting differences between experiments.

How many times was each experiment independently repeated?

What is the source of the reported error bars? Are they errors of the fit? Standard deviations?

Validity of the findings

see below

Additional comments

This is a very clearly written manuscript that reports the reconciliation of in vitro binding affinities with in vivo repression for LacI under both non-inducing and inducing conditions. No one has attempted to do this for the full functional cycle of LacI.

The author has paid careful attention to a number of controls and to various model/experimental parameters. They have measured in vivo protein and DNA concentrations, which is very important and difficult parameter for such a work.

The paper has two results about which I am not yet wholly convinced by the author’s interpretation.

1. Results show that the in vitro affinity for LacI binding to DNA is weaker binding than previously published. The authors attributed all of the difference to technique used to monitor binding, which could play a role. The other techniques have known artifacts.

However, another important consideration is buffer differences. I cannot find the buffer conditions of the reported experiments stated anywhere in the manuscript. (Whether or not this contributes, it should be included). In vitro DNA binding is particularly sensitive to buffer conditions.

Second how long is the DNA used in the in vitro experiment? Are sufficient quantities present to diminish the apparent affinity for DNA via nonspecific binding?

Third, did the authors measure the LacI activity (fraction of active protein) used in their in vitro preparation under stoichiometric conditions? It is not unusual to have a significant amount of “dead” LacI after purification. The mCHerry tag could likely still remain active when the LacI DNA binding is dead. Along the lines of the missing buffer conditions – was BSA used to prevent the LacI from sticking to plasticware at low protein concentrations? That would give the artifact of diminished DNA binding.

2. In vivo/in vitro disagreement: The in vitro parameters predict a system that is more leaky than is measured.
The author shows that molecular crowding can account for this difference. However, as noted above, buffer conditions were not discussed. Nonspecific binding was ruled out because it should have the opposite outcome; would this still be true if the in vitro DNA binding affinity agreed with all other stronger measurements?

The modeling for molecular crowding accounts for the discrpancy just as conclusively as Tungtur et al account for their discrepancy – by reasonable estimation of probable, confounding factors. A more interesting note is that the disagreement of the current study is OPPOSITE to the disagreement found by Tungtur et al. What could give rise to the opposite experimental results? What differs in the experimental setup of the two manuscripts?

More importantly, the authors use an incorrect assumption that could influence the model, especially under the induced conditions (which shows larger discrepancy in Figure 5). When LacI binds DNA, the binding affinity does not go to zero, as is incorporated in the current model. Instead, the LacI-IPTG binding to DNA has been measured to be 6.3 μM^-1 (Daly and Matthews, Allosteric Regulation of Inducer and Operator Binding to the Lactose Repressor. Biochemistry 1986, 25, 5479-5484). While this value certainly could differ with solution conditions or with technique, it is clearly a much larger value than that used in the model.

If a larger value is incorporated into their model, does this resolve the discrepancy without invoking molecular crowding? The parameter need not be floated during the fit – the author could include it as a fixed parameter over a range of values.

Reviewer 3 ·

Basic reporting

The article by M. Sochor is well written and well structured. The figures are of good quality and illustrate major findings of this work. Relevant literature is appropriately referenced and the introduction contains sufficient information to be able to appreciate the interest of this work in a larger context.

Experimental design

The work presented here is original and relevant. The aims are clearly indicated and the methods used are described in great detail.

Validity of the findings

The experiments are carefully planned and performed. The experimental results generally support the major conclusions.

Additional comments

The work presented in this manuscript is interesting, also in the context of application of modified operator/repressor couples as regulatory switches in both E. coli and heterologous hosts. However, as the operator and repressor used in this work differ significantly from the natural E. coli Lac repressor-operator system this should be very clearly indicated, especially when comparisons with previous results are made. Line 312 a comparison with the work of B. Müller-Hill wis made. Was this previous performed with a single or tripartite operator? Indicate the correspondence or deviation from the structure used in this work.
Also, it is not entirely clear why the template plasmid DNA for in vitro transcription is linearized. Is this to generate run-off transcripts? Linear and supercoiled plasmid DNA may behave somewhat different in in vitro transcription and repressor binding. Indicate this in Materials and methods section. It is indicated (line 146) that the operator O1 was put after the T7 promoter. Does the lac repressor then act as a roadblock or is there still interference with binding of RNA polymerase? Could the author comment on this point.
Line 360-368: the author enumerates a number of differences between the system used in this work and the lac operon. A difference that should be added to this list is the different position of the sole lac repressor binding site used here and of the natural tripartite operator with respect to the RNA polymerase binding site.

---

## Round 0.2 · accepted · Accept

Thanks for making the revision.